# Deep learning approaches for quantitative and qualitative assessment of cervical vertebral maturation staging systems

Abbas Ahmed Abdulqader[1☯], Fulin Jiang[2☯], Bushra Sufyan Almaqrami[3], Fangyuan Cheng[4], Jinghong Yu[2], Yong Qiu[2], Juan Li[1*]

1 State Key Laboratory of Oral Diseases and National Center for Stomatology and National Clinical Research Center for Oral Diseases, West China Hospital of Stomatology, Sichuan University, Chengdu, China, 2 Chongqing University Three Gorges Hospital, Chongqing, China, 3 Department of Orthodontics, Ningbo Dental Hospital, Ningbo, Zhejiang, China, 4 Chengdu Boltzmann Intelligence Technology Co., Ltd, Chengdu, China

☯ These authors contributed equally to this work.
* lijuan@scu.edu.cn

## Abstract

To investigate the potential of artificial intelligence (AI) in Cervical Vertebral Maturation (CVM) staging, we developed and compared AI-based qualitative CVM and AI-based quantitative QCVM methods. A dataset of 3,600 lateral cephalometric images from 6 medical centers was divided into training, validation, and testing sets in an 8:1:1 ratio. The QCVM approach categorized images into six stages (QCVM I–IV) based on measurements from 13 cervical vertebral landmarks, while the qualitative method identified six stages (CS1–CS6) through morphological assessment of three cervical vertebrae. Statistical analyses evaluated the methods' performance, including the Pearson correlation coefficient, mean square error (MSE), success detection rate (SDR), precision-recall metrics, and the F1 score. For landmark prediction, our AI model demonstrated remarkable performance, achieving an SDR (error threshold of ≤ 1.0 mm) of 97.14% and with the mean prediction error across thirteen landmarks ranging narrowly from 0.17 to 0.55 mm. Based on the AI-predicted landmarks, the cervical vertebral measurements showed strong agreement with orthodontists, as indicated by a Pearson correlation coefficient of 0.98 and an MSE of 0.004. Besides, the CVM method attained an overall classification accuracy of 71.11%, while the QCVM method showed a higher accuracy of 78.33%. These findings suggest that the AI-based quantitative QCVM method offers superior performance, with higher agreement rates and classification accuracy compared to the AI-based qualitative CVM approach, indicating the fully automated QCVM model could give orthodontists a powerful tool to enhance cervical vertebral maturation staging.

**Data availability statement:** We have made all the code available at "https://github.com/ortho2024/QCVM-CVM" that can be used to replicate this study. The data underlying the findings of this study cannot be made publicly available due to ethical and legal restrictions. The datasets can't be publicly deposited because they would compromise patient privacy and violate the protocol approved by the research ethics board of West China Hospital of Stomatology, Sichuan University. However, the data are available from the corresponding author upon reasonable request, subject to approval by the ethics committee and compliance with data protection regulations. Researchers interested in accessing the data may contact [Juan Li] at [E-mail: lijuan@scu.edu.cn] to submit their request.

**Funding:** This work was supported by the Science and Health Joint Medical Research Project Fund of Chongqing (Grant No. 20250QNXM023), the National Key Research and Development Program of China (Grant No. 2024YFC2510704), and the Angelalign Scientific Research Fund (Grant No. SDTS21–4–01). The funders had no role in study design, data collection and analysis, decision to publish, or preparation of the manuscript.

**Competing interests:** The authors have declared that no competing interests exist.

## 1. Introduction

Cervical vertebral maturation (CVM) is a crucial parameter in orthodontics and growth assessment, used to determine the skeletal age and predict future growth spurts in adolescents [1]. Accurate assessment of CVM stages is essential for treatment planning and prognosis in orthodontic care. Initially, this assessment relied on qualitative methods, where experts visually analyze vertebral shapes and patterns on lateral cephalometric radiographs, and quantitative methods, which involve precise measurements of vertebral dimensions [2]. Despite their utility, these traditional approaches can be time-consuming, subjective, and prone to variability among primary orthodontists [3].

The evaluation of cervical vertebral maturity (CVM) was initially introduced by Lamparski et al. [4], advocated lateral cephalometric analysis as a valid alternative to hand-wrist radiographs for assessing growth maturation. Subsequently, the methodology evolved through significant refinements, including a reduction in the vertebrae assessed from six to three and a detailed characterization of each stage based on the shape of the three cervical vertebra [5]. Recent meta-analyses have supported the potential to replace wrist radiographs, thus reducing patients' exposure to additional X-rays [6]. Nonetheless, the reproducibility of this method has been the subject of debate, evidenced by divergent findings across various studies [7].

Building upon this, further improvements were made to the Quantitative Cervical Vertebral Maturation (QCVM) analysis developed by Baccetti et al. [8], aimed for simplification and broader applicability, aligning with the recommendations by Hassel and Farman [5]. To address these concerns, recent propositions include analytical techniques that employ points, angles, and equations to enhance the precision of stage classification [9].

Recent advancements in artificial intelligence (AI) have opened new avenues for enhancing the accuracy and efficiency of CVM assessment. Deep learning models, particularly convolutional neural networks (CNNs), have demonstrated remarkable success in medical image analysis by automatically extracting relevant features and patterns from radiographic images. These models can be trained to perform qualitative and quantitative assessments, potentially reducing the reliance on expert manual scoring and minimizing the time required for analysis.

The accuracy of cervical vertebral maturation using deep learning in previous studies has shown varying levels of accuracy, ranging from 0.59 to 0.71 [3,10–14]. These studies often emphasize qualitative assessments with limited quantitative measurements [15]. Recent studies demonstrated a high accuracy classification rate that consolidates two stages to be a three-stage scenario instead of six classifications [14,16].Additionally, a meta-analysis systematically evaluates the performance of artificial intelligence (AI) models in assessing cervical vertebral maturation (CVM) from radiographs, highlighting their potential to enhance accuracy and reliability compared to traditional clinical methods [17].

In this study, we investigate the potential of deep learning to enhance the accuracy and efficiency of CVM assessment. We conduct a comparative analysis of qualitative

CVM and quantitative QCVM methods when augmented with AI. Thereby supporting better-informed decisions on the timing of orthopedic treatment.

**Our main contributions are as follows:**

1) We present a large, evenly distributed dataset encompassing all stages of cervical vertebral maturation collected from multiple centers to ensure diversity and generalizability.

2) As the first comparative study of AI based on QCVM and CVM, this work provides critical insights into the effectiveness and clinical practicality of various cervical maturation staging methods.

3) We developed deep learning models by employing six maturation stages assessment methods for both quantitative and qualitative analysis (as depicted in Fig 1). The CVM staging adheres to the original framework proposed by Baccetti et al. [8] This integration allows us to bridge classical methodologies with modern AI-driven approaches effectively.

## 2. Materials and methods

### 2.1. Ethical approval

Data were accessed for research between 20 May 2019 and 30 June 2021. An initial dataset of 6000 lateral cephalometric radiographs was randomly collected from six orthodontic clinics. After applying exclusion criteria, 2400

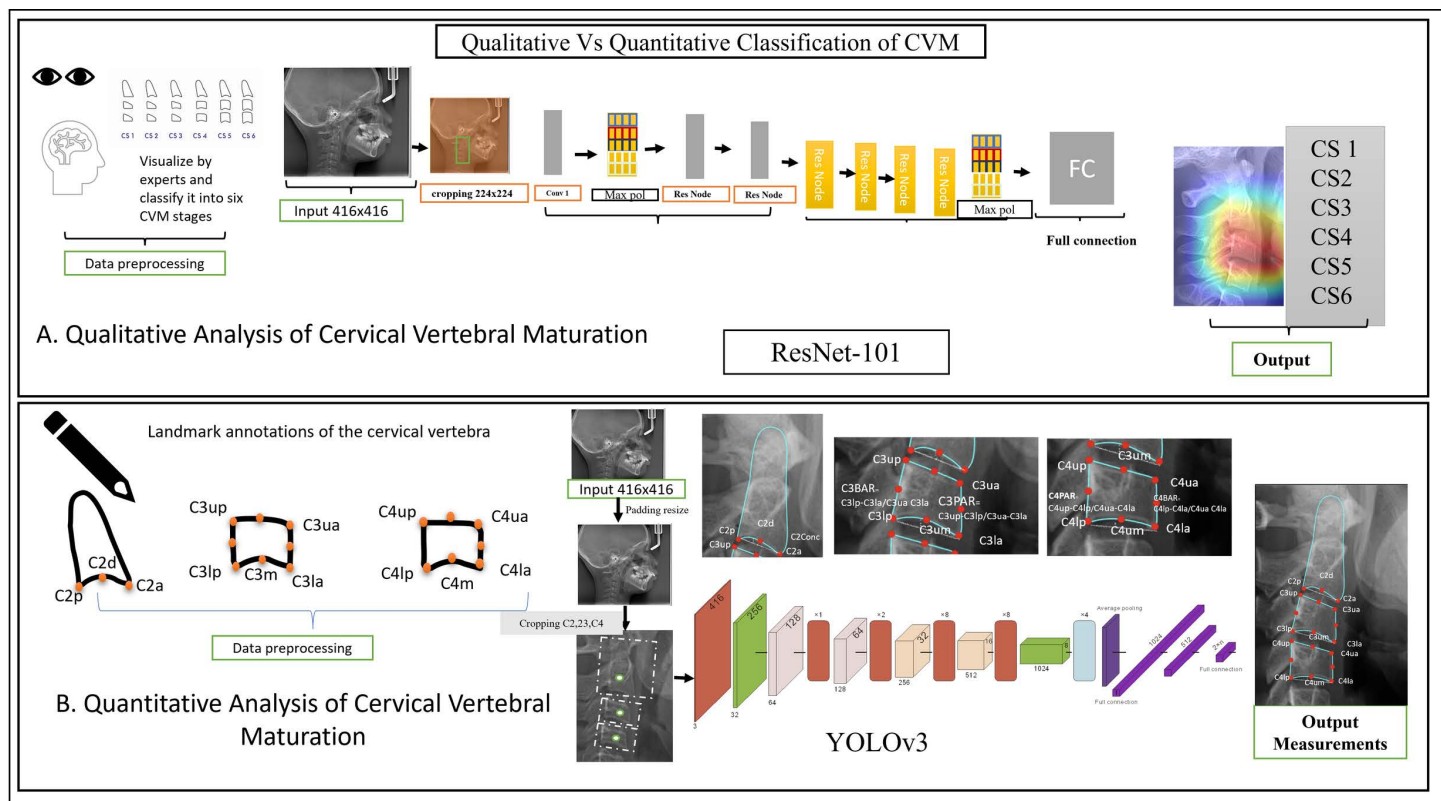

**Fig 1. Workflows of our study (Comparison of Qualitative and Quantitative CVM Analysis).** (a) Qualitative analysis uses ResNet-101 to classify images into six CVM stages (CS1–CS6) based on vertebral morphology. (b) Quantitative analysis applies YOLOv3 to detect anatomical landmarks and compute measurements for staging (QCVM I–VI).

cephalograms were excluded, resulting in a final dataset of 3600 images. Exclusions were based on the following criteria: (1) incomplete visualization of all three cervical vertebrae, (2) balancing the sample distribution across Cervical Stage (CS) classifications (CS1 to CS6) to ensure equal representation, and (3) standardizing the dataset to maintain uniform sample sizes across all stages for consistent analysis. All study procedures adhered to the ethical guidelines of the Helsinki Declaration, and the protocol received approval from the Institutional Review Board (approval number WCHSIRB-D-2019–120).

## 2.2. Data collection and preprocessing

**2.2.1. Data collection.** The final dataset comprises 3600 lateral cephalometric images, exhibiting a range of image qualities and diverse vertebral shapes and positions, making it ideal for comprehensive analysis. Participants were selected based on specific criteria: absence of bone growth impairments, no systemic diseases or developmental delays, and no congenital or acquired anomalies in the craniofacial region. All lateral cephalometric radiographs are needed to display the second (C2), third (C3), and fourth (C4) cervical vertebrae. Initially captured in DICOM format, these radiographs were later converted to PNG format for analysis. The image resolutions ranged from 416 to 2200 pixels in width and from 416 to 2500 pixels in height. After the data annotation, the radiographs were randomly split into training, validation, and test sets in an 8:1:1 ratio for each method and subdivision at every stage.

**2.2.2. Data preprocessing.**

1) Qualitative Annotation

The final data set consisted of 3600 radiographs and was even distributed at six classification stages. Four distinct shapes were identified: trapezoidal, rectangular horizontal, square, and rectangular vertical. These shapes were assigned to specific stages of vertebral maturation based on the guidelines established by McNamara et al. in a recent user guide [18]. As shown in Fig 1a and S1 Table. Three orthodontists carried out this classification process: two junior orthodontic residents with over five years of annotation experience and one senior professor of orthodontics. Before the annotation process, two orthodontists received training and evaluation from a senior orthodontist professor. For reliability assessment, 10% of each classification category was sampled. The results showed that Cohen's Kappa between the senior orthodontist and Junior Orthodontist 1 was 0.78, between the senior and Junior Orthodontist 2 was 0.84, and between the two juniors was 0.85. The overall Fleiss' Kappa value was 0.82, indicating a high level of inter-rater agreement.

2) Quantitative Annotation (landmark annotation)

Quantitative annotation was conducted by manually localizing landmarks on 3,600 lateral cephalometric radiographs— thirteen specific landmarks defined by Baccetti et al. [8]. Were annotated by two junior orthodontists. The staging ground truth was established using the McNamara et al. user guide [19].

Fig 1b illustrate the localization of the landmarks, while their definitions are provided in S2 Table. Specifically, the second cervical vertebra (C2) includes three landmarks, the third vertebra (C3) includes five, and the fourth vertebra (C4) includes five as well. These landmarks constitute the measurement framework for the Quantitative Cervical Vertebral Maturation (QCVM) method, which is further detailed in S3 Table.

These landmarks form the basis of the QCVM measurement system, which comprises seven indicators focused on morphological assessment: C2Conc, C3Conc, and C4Conc measure the depth of concavity at the lower border of each respective vertebra; C3PAR and C4PAR represent the ratio of posterior to anterior vertebral body heights; and C3BAR and C4BAR quantify the ratio between the base length and anterior height of the vertebral bodies. Together, these measurements provide a standardized framework for the QCVM staging system, with information subsequently extracted through qualitative analysis of the corresponding cephalograms.

To ensure the landmark annotation accuracy and consistence, we trained the two junior orthodontists and selected 100 cephalograms to test the annotation differences. The average SDR across 13 landmarks was 97% at a 1 mm error threshold, confirming consistent landmark annotation.

3) Data augmentation

Our study used image augmentation techniques to simulate clinical conditions to improve the model's performance. We chose several geometric and photometric transformations relevant to clinical settings, including contrast, denoising, inverting, solarizing, and rotation noise addition, as shown in Fig 2. These transformations were randomly applied within predefined limits for the maximum extent of modification, ultimately resulting in approximately 6480 images.

## 2.3. Neural network architecture and training details

### 2.3.1. Qualitative CVMS determination (CVM).
The standard qualitative method for estimating cervical vertebral maturation involves two sets of procedures, as depicted in Fig 1a and Fig 3. The analysis begins with the neural network using a ResNet-101 architecture [20]. This process involves initially padding and cropping the vertebra to a size of 416 x 416 pixels. The initial phase of the analysis classifies the vertebrae into C2, C3, and C4, each subsequently cropped into a square to ensure precise analysis. A heat-mapping process is applied to each classified region to enhance the accuracy, highlighting areas of significant morphological change critical for accurate maturation assessment. The second procedure leverages the neural network to discern the characteristics of these three vertebrae, establishing ground truth based on the concavity at the lower borders of the vertebral bodies of C2, C3, and C4, along with identifying four distinct shape differences. The output layer delineates six stages of maturation, each corresponding to specific developmental phases, determined through the combined assessments of concavity, shape, and heat map analyses.

### 2.3.2. Quantitative method landmarks identification.
To accurately assess the orientation and structure of cervical vertebrae, we designed a neural network architecture tailored to localize specific anatomical landmarks on the C2, C3, and C4 vertebrae. The network is structured to detect three key landmarks on the C2 vertebra, five on the C3 vertebra, and five on the C4 vertebra. The network architecture used YOLO v3 architecture [21], including the input, hidden, and output layers. The input layer accepts images with resolutions ranging from 416 to 2200 pixels in width and 416–2500 pixels in height and resizes to 416*416 pixels. The hidden layer begins with convolutional layers, followed

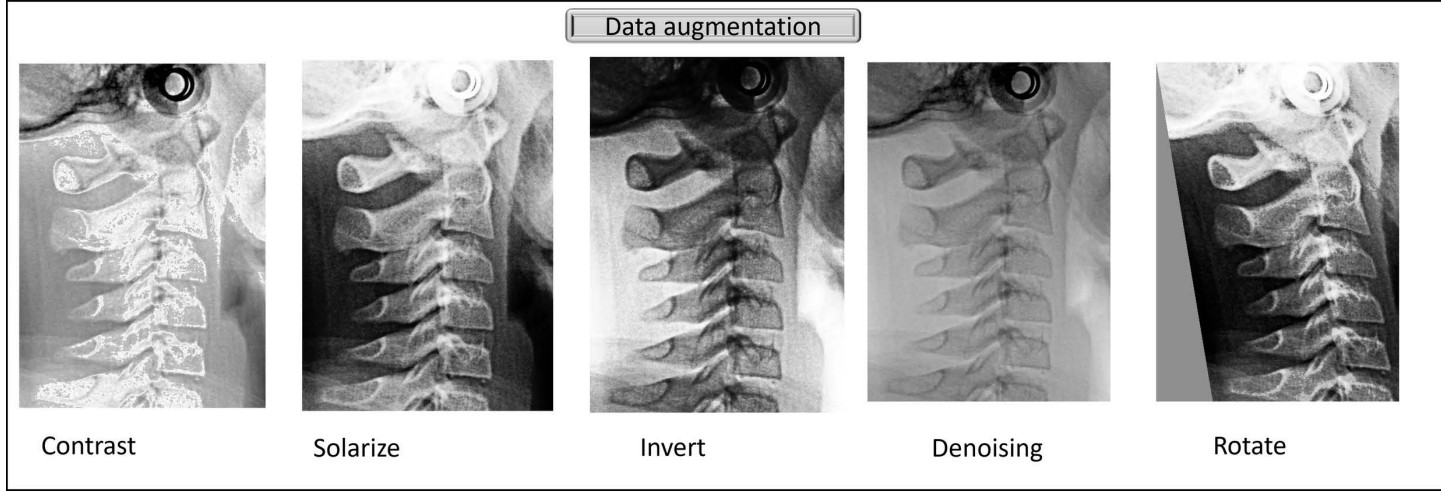

**Fig 2. Examples of data augmentation techniques applied to lateral cephalometric radiographs.**

by batch normalization and max pooling, to extract hierarchical features from the input images. Specifically, the hidden layer comprises two major blocks: the encoder and decoder blocks. The encoder block is mainly responsible for feature extraction and downsampling, which consists of two initial convolutional layers with 32 and 64 filters, respectively, followed by two additional convolutional layers with 128 filters each. The decoder block decodes the learned features to produce the required output, which consists of 2 fully connected layers with 512 and 256 units, respectively, each followed by dropout for regularization. The output layer consists of 12 units, corresponding to the total number of landmarks (3 for C2, 5 for C3, and 5 for C4), with a linear activation function to predict the precise coordinates of each landmark. This architecture is designed to localize the specified landmarks, facilitating detailed analysis and assessment of the cervical vertebrae.

**2.3.3. Quantitative cervical vertebral maturation staging (QCVM).** The cervical vertebrae's quantitative analysis metrics were determined using 13 landmarks outlined by Baccetti et al. [8]. Built on these landmarks, the measurements relate to the anatomical features of the cervical vertebrae C2, C3, and C4, particularly examining concavity depth and proportional dimensions. C2Conc, C3Conc, and C4Conc measure the concavity depth at the lower borders of C2, C3, and C4, respectively, by gauging the distance from a baseline to the deepest point on each vertebra's lower border. Ratios such as C3PAR and C4PAR assess the relative heights of the posterior and anterior walls of C3 and C4. Meanwhile, C3BAR and C4BAR determine the ratios of the base length to the anterior height of these vertebrae.

Furthermore, five key measurements, along with the established golden standard for Cervical Vertebral Maturation (CVM), were mapped to the corresponding ID of each image data set. These datasets were then employed to train the

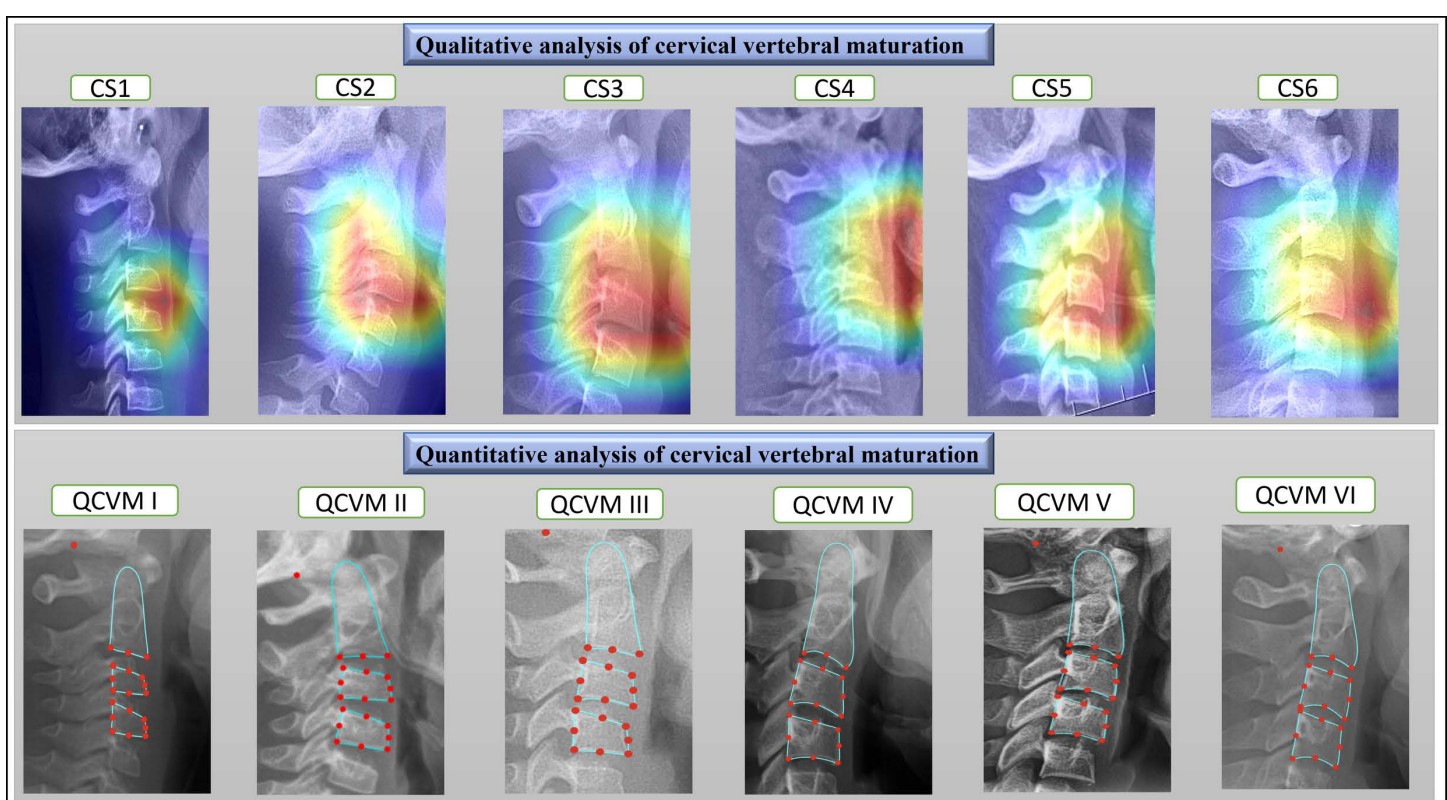

**Fig 3. Qualitative and quantitative analyses of cervical vertebral maturation (CVM).** The top panel shows heatmaps for six maturation stages (CS1–CS6), while the bottom panel presents landmark-based annotations for the Quantitative CVM framework (QCVM I–QCVM VI).

classification model using the CatBoost algorithm [22], which achieved a total accuracy of 0.74. This implementation showcases the effectiveness of CatBoost in handling complex feature interactions in a highly dimensional space, as shown in Fig 4.

**2.3.4. Implementation details.** The Adam optimizer was used with an initial learning rate of 0.001 divided by 10 every 50 epochs. Both networks were trained 500 epochs. It took about 12 hours to train the ROI detection network, 14 hours for the localization network, and two hours for Full connection on a single NVIDIA GeForce RTX 3090 GPU.

## 2.4. Evaluation metrics

**2.4.1. Qualitative CVM staging performance.** The evaluation of 360 test images involved comparing the final assessments of cervical vertebral maturation (CVM) stages, ranging from CS1 to CS6, between a deep learning model trained for standard qualitative assessment and human assessments that were conducted by three expert orthodontists. The accuracy of these classifications was evaluated using confusion matrices, recall precision and F1 score. This comprehensive evaluation approach allowed for a detailed comparison of the model's performance against expert human assessments, providing insights into the model's reliability and areas for potential improvement, as depicted in Table 1 and Fig 5.

To evaluate the agreement among the three annotators in qualitative cervical staging (CS) assessment, we calculated Cohen's Kappa and Fleiss' Kappa to assess inter-rater reliability.

**2.4.2. QCVM metrics.** The evaluation metrics for QCVM are conducted in three steps: firstly, by assessing landmark positional differences; secondly, through measurement agreements; and thirdly, by classifying QCVM staging.

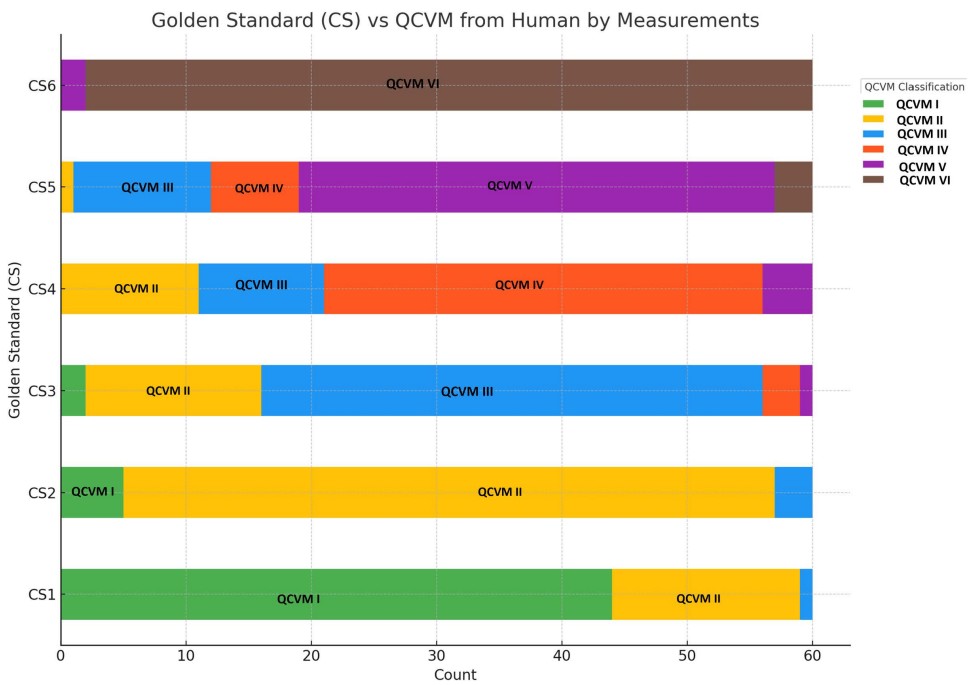

**Fig 4. Comparative analysis of the Golden Standard (CS) qualitative classification and QCVM quantitative classification applied to a dataset of 360 samples.** The stacked bar chart illustrates the correspondence between qualitative cervical staging (CS1–CS6) and quantitative measurements (QCVM1–QCVM6), highlighting areas of agreement and discrepancy between the two methods.

**Table 1. The CVM qualitative analysis confusion metrics of every class of 360 test sample.**

| Class | True Positives (TP) | False Positives (FP) | False Negatives (FN) | Precision | Recall | F1 Score | Accuracy |
|---|---|---|---|---|---|---|---|
| CS1 | 43 | 4 | 17 | 0.91 | 0.71 | 0.80 | |
| CS2 | 40 | 33 | 20 | 0.54 | 0.66 | 0.60 | |
| CS3 | 34 | 25 | 26 | 0.57 | 0.56 | 0.57 | |
| CS4 | 39 | 21 | 21 | 0.65 | 0.65 | 0.65 | |
| CS5 | 49 | 19 | 11 | 0.72 | 0.81 | 0.76 | |
| CS6 | 51 | 2 | 9 | 0.96 | 0.85 | 0.90 | |
| Overall | 256 | 104 | 104 | | | | 71.11% |

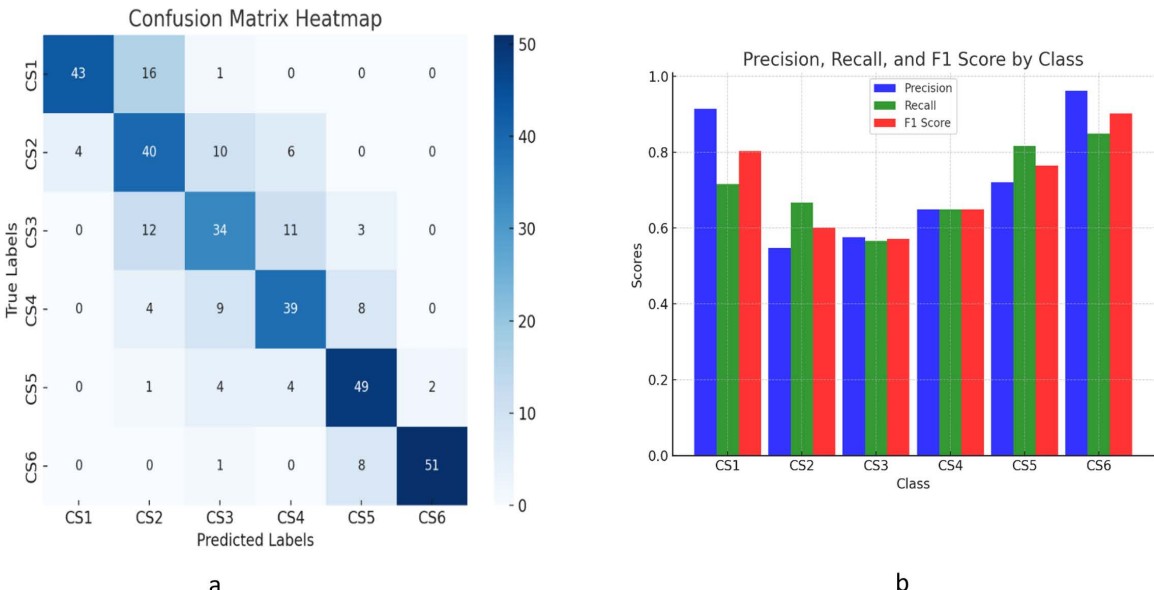

a

b

**Fig 5. (a) Confusion matrix heatmap illustrating the performance of qualitative cervical vertebral maturation (CVM) classification, showing true versus predicted labels distribution.** (b) Bar chart depicting precision, recall, and F1-score for each CVM class, highlighting the classification performance across different stages.

1) Measuring the position of landmarks

In the accuracy assessment of 360 test images for AI landmark identification compared to human annotations, both the AI system and expert annotators determine the coordinates (X, Y) of designated landmarks on the images. Using the same coordinate system and scale, the Euclidean distance between corresponding landmarks identified by both methods is calculated. A threshold of 2 mm is established as the maximum allowable distance difference for the landmarks to be considered accurate.

2) Measurement agreements

Seven key measurement indicators were analyzed to evaluate the agreement between human and AI assessments. We employed the Pearson correlation coefficient to assess the linear relationship between the two sets of measurements, and the Mean Squared Error (MSE) was used to quantify the differences between the values determined by humans and AI. These statistical tools help in understanding the consistency and accuracy of AI relative to human evaluators, providing a robust framework for comparison.

3) Quantitative QCVM Staging Performance

Seven key measurements, derived from both human and AI annotations, were employed to classify 360 images, spanning QCVM I through QCVM VI, using a previously trained CatBoost algorithm. The effectiveness of this classification was assessed through precision, recall, and the F1 score, with the overall accuracy also being measured. These performance metrics are shown in Table 5 and Fig 6.

## 3. Result

The analysis of the 360 cephalometric test images provided significant insights into the performance of both the AI and manual classification in assessing cervical vertebral maturation (CVM) stages.

### 3.1. Qualitative classification metrics

The confusion matrix analysis for the qualitative CVM staging indicated that the AI model achieved a notable level of accuracy. Precision rates varied across different classes, with the highest being 0.96 in CS6 and the lowest at 0.54 in CS2. Recall rates also showed variation, with CS5 achieving the highest at 0.81 and CS3 the lowest at 0.56. The overall accuracy of the model was 71.11%, and the overall precision was calculated at 0.687, as shown in Table 1. These results demonstrate that the AI model is effectively classifying the stages of cervical vertebral maturation, closely aligning with the manual assessments conducted by orthodontists.

### 3.2. Landmarks position difference

The positional differences between AI and human annotations for the test data images are shown in Table 2. The analysis reveals that, on average, discrepancies between the AI and human measurements are minimal, as evidenced by the mean and standard deviation (SD) values. For instance, landmarks such as C3up and C4um exhibit relatively low mean differences at 0.29 mm and 0.15 mm, respectively, suggesting close alignment between AI and human annotators.

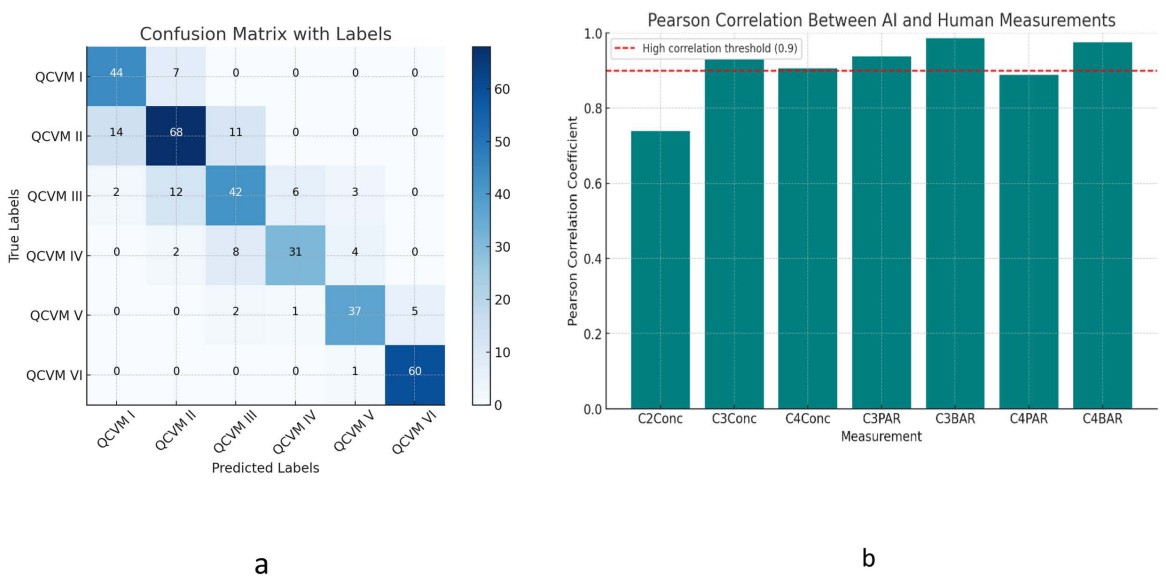

a                                           b

**Fig 6. (a) Confusion matrix heatmap illustrating the performance of quantitative cervical vertebral maturation (QCVM) classification, showing the distribution of true versus predicted labels.** (b) Pearson correlation coefficients between AI-based QCVM measurements and human annotations, demonstrating strong correlations across most measurements, with a high correlation threshold (0.9) indicated by the red dashed line.

**Table 2. Positional difference of landmarks between human and AI.** A value of 0 indicates perfect agreement, while values less than 2 mm are considered clinically acceptable. CL (Confidence Interval) represents the 95% confidence range around the mean value, indicating the reliability of the estimate.

| Landmarks | Mean | SD | Median | Min | Max | 95%CI |
|---|---|---|---|---|---|---|
| C2p (mm) | 0.55 | 0.37 | 0.50 | 0 | 1.91 | 0.51~0.59 |
| C2m (mm) | 0.41 | 0.33 | 0.35 | 0 | 1.42 | 0.38~0.45 |
| C2a (mm) | 0.51 | 0.42 | 0.40 | 0 | 2.43 | 0.46~0.55 |
| C3up (mm) | 0.29 | 0.29 | 0.20 | 0 | 1.89 | 0.26~0.32 |
| C3ua (mm) | 0.20 | 0.19 | 0.19 | 0 | 1.55 | 0.18~0.22 |
| C3lp (mm) | 0.32 | 0.29 | 0.26 | 0 | 1.59 | 0.29~0.35 |
| C3la (mm) | 0.29 | 0.29 | 0.21 | 0 | 2.06 | 0.26~0.32 |
| C3m (mm) | 0.17 | 0.18 | 0.13 | 0 | 1.25 | 0.32~0.39 |
| C4up (mm) | 0.35 | 0.32 | 0.26 | 0 | 1.47 | 0.13~0.17 |
| C4ua (mm) | 0.21 | 0.21 | 0.192 | 0 | 1.07 | 0.21~0.26 |
| C4um (mm) | 0.15 | 0.17 | 0.12 | 0 | 1.57 | 0.30~0.36 |
| C4lp (mm) | 0.33 | 0.28 | 0.27 | 0 | 2.09 | 0.26~0.32 |
| C4la (mm) | 0.29 | 0.281 | 0.21 | 0 | 2.37 | 0.15~0.19 |
| C4m (mm) | 0.21 | 0.19 | 0.18 | 0 | 1.06 | 0.18~0.23 |

However, despite generally minor differences across most landmarks, occasional outliers occur, such as the higher maximum deviations noted at C2a and C4la. These instances are exceptions rather than indicative of the overall accuracy of the AI model.

The success detection rates (SDRs) across 13 landmarks at precision thresholds of ≤0.5 mm, ≤1.0 mm, and ≤1.5 mm. Most landmarks achieved high detection rates at the ≤1.0 mm (85.52–100%) and ≤1.5 mm (97.21–100%) thresholds, as illustrated in Table 3. However, performance at the more stringent ≤0.5 mm threshold was more variable, ranging from 49.86% for C2p to 97.49% for C3um. Posterior landmarks—particularly C2p and C2a—showed the lowest sub-millimeter accuracy, while medial and upper landmarks (e.g., C3um, C4um, C3ld, C4ld) consistently demonstrated high precision. These findings indicate that the system performs reliably for most landmarks within clinically acceptable tolerances (≤1.5 mm).

### 3.3. Measurement agreements

The agreements between AI and human measurements are detailed in Table 4. This comparison reveals a strong alignment in the assessments of cervical vertebral metrics. For concavity measurements such as C2Conc, C3Conc, and C4Conc, both AI and human annotations demonstrate closely matched means and standard deviations, with Pearson Correlation scores indicating robust correlations—0.73, 0.93 and 0.90, respectively. These high correlations, along with low Mean Squared Errors (MSE), underscore the precision of the AI in mirroring human evaluative accuracy.

Additionally, the ratio measurements, including C3PAR, C3BAR, C4PAR, and C4BAR, exhibit exceptionally high consistency between AI and human results. The Pearson Correlations for C3BAR and C4BAR are notably high at 0.98 and 0.97, respectively, supported by minimal MSE values, highlighting an almost perfect agreement.

### 3.4. Quantitative classification metrics

The detailed classification results for cervical vertebral maturation stages, ranging from QCVM I to QCVM VI, are comprehensively outlined in Table 5. This table illustrates varied performance metrics, including precision, recall, and F1 scores for each stage. For example, QCVM VI shows exemplary performance with a precision of 0.92 and a recall of 0.98, resulting in an F1 score of 0.95. The overall accuracy across all stages is noted as 78.33%, indicating the AI

**Table 3. Success Detection Rate (SDR) of AI predictions within the given distance from human landmarks.**

| Landmarks | SDR | | |
|---|---|---|---|
| | ≤ 0.5 mm (%) | ≤ 1.0 mm (%) | ≤ 1.5 mm (%) |
| C2p | 49.86 | 87.19 | 98.33 |
| C2d | 64.35 | 93.87 | 100.00 |
| C2a | 57.94 | 85.52 | 97.21 |
| C3up | 79.39 | 97.49 | 99.72 |
| C3ua | 91.64 | 99.72 | 99.72 |
| C3lp | 75.49 | 97.77 | 99.44 |
| C3la | 79.94 | 96.94 | 99.44 |
| C4up | 70.75 | 93.87 | 100.00 |
| C4um | 95.54 | 99.72 | 99.72 |
| C4am | 87.19 | 99.44 | 100.00 |
| C4lp | 73.82 | 98.89 | 99.72 |
| C4la | 80.50 | 98.33 | 99.44 |
| C3ld | 93.59 | 99.72 | 100.00 |
| C4ld | 92.20 | 99.72 | 100.00 |
| C3pm | 86.91 | 98.89 | 100.00 |
| C3am | 90.81 | 99.72 | 100.00 |
| C3um | 97.49 | 100.00 | 100.00 |
| C4pm | 87.47 | 99.16 | 100.00 |
| C4ua | 91.36 | 99.72 | 100.00 |

**Table 4. Comparison of quantitative measurements of orthodontists and deep learning.**

| Measurement | AI | | | Orthodontists | | | Pearson Correlation | MSE |
|---|---|---|---|---|---|---|---|---|
| | Mean | Sd | 95%CI | Mean | Sd | 95%CI | | |
| C2Conc mm | 0.64 | 0.59 | 0.58~0.70 | 0.74 | 0.58 | 0.80~0.73 | 0.73 | 0.19 |
| C3Conc mm | 0.7 | 0.64 | 0.63~0.76 | 0.64 | 0.6 | 0.71~0.93 | 0.93 | 0.058 |
| C4Conc mm | 0.58 | 0.54 | 0.52~0.63 | 0.58 | 0.53 | 0.63~0.9 | 0.9 | 0.054 |
| C3PAR (ratio) | 1.34 | 0.18 | 1.32~1.36 | 1.33 | 0.18 | 1.35~0.93 | 0.93 | 0.004 |
| C3BAR (ratio) | 1.68 | 0.48 | 1.63~1.73 | 1.68 | 0.47 | 1.73~0.98 | 0.98 | 0.006 |
| C4PAR (ratio) | 1.4 | 0.16 | 1.38~1.41 | 1.4 | 0.19 | 1.42~0.88 | 0.88 | 0.007 |
| C4BAR (ratio) | 1.71 | 0.44 | 1.66~1.76 | 1.73 | 0.45 | 1.78~0.97 | 0.97 | 0.01 |

model's robust capability in effectively classifying different stages of cervical vertebral maturation, with extreme accuracy in the advanced stages.

## 3.5. Efficiency comparison

The timing records of 360 test data between expert classification and the trained model revealed that the AI model processed and classified the images in just 7 minutes for both CVMS and QCVM, whereas junior orthodontists took an average of 53 minutes in CVM assessment and 347.64 minutes in landmark localization of QCVM. This significant difference in processing time suggests that the AI model can offer much faster diagnostic times in clinical settings, enhancing efficiency without compromising accuracy.

**Table 5. QCVM quantitative analysis confusion metrics of every class of 360 test sample.**

| Class | TP | FP | FN | Precision | Recall | F1 Score | Accuracy |
|---|---|---|---|---|---|---|---|
| QCVM I | 44 | 16 | 7 | 0.733 | 0.86 | 0.79 | |
| QCVM II | 68 | 21 | 25 | 0.76 | 0.73 | 0.74 | |
| QCVM III | 42 | 21 | 23 | 0.66 | 0.64 | 0.65 | |
| QCVM IV | 31 | 7 | 14 | 0.81 | 0.68 | 0.74 | |
| QCVM V | 37 | 8 | 8 | 0.82 | 0.82 | 0.82 | |
| QCVM VI | 60 | 5 | 1 | 0.92 | 0.98 | 0.95 | |
| Total Accuracy | | | | | | | 78.33 |

## 4. Discussion

This study offers a clinical perspective on comparing two deep learning-based methods for CVM estimation—a critical area in orthodontics that influences treatment planning and prognosis.

The results of this study are clinically significant, particularly when considering the daily challenges faced by orthodontists in accurately determining skeletal maturity. Previous research [3,12,14] has primarily focused on qualitative CVM assessments, which have shown variable accuracy rates. This variability is especially pertinent in clinical practice, where consistent and reliable assessments are essential for effective treatment planning. Our study, by comparing quantitative and qualitative methods, provides valuable insights into improving these assessments. The qualitative assessment achieved an accuracy rate of 0.71, aligning with the result from a previous study [12]. This contrasts with some studies [3,11] reporting lower accuracy rates in qualitative assessments, though higher accuracy has been noted when grouping six stages into three broad categories [23]. In comparison, the quantitative method in our study demonstrated a higher accuracy rate of 0.78, using an automated, consistent methodology. In contrast to our study, a recent automated quantitative analysis involving a sample of 560 female cases was conducted [15]. These cases were classified into four stages, thereby exhibiting a high accuracy rate.

Khazaei et al. [24], and Radwan et al. [23], simplified the six CVM stages into three grouped categories by combining adjacent stages. Despite achieving high classification accuracy, their method faces limitations in clinical settings, particularly in precisely predicting craniofacial growth as the original description [5]. Contrarily, most AI approaches, including those by Radwan et al. [23], Rahimi et al. [14], and Kim et al. [11], utilize the segmentation technique for the CVM classification. These methods are prevalent in medical image analysis but encounter obstacles in CVM classification due to the intensive computational requirements and the need for exact annotations.

Clinically, the comparison between the qualitative and quantitative methods also raises important considerations. While the qualitative method provides a more nuanced understanding of vertebral morphology, which can be beneficial in some clinical contexts, it also introduces variability due to its subjective nature. This variability could impact diagnosis decisions, especially in borderline cases where precise staging is critical. The quantitative method, on the other hand, offers a more consistent and reliable approach, which could lead to more predictable treatment expectations.

When comparing the classification methods for cervical vertebral maturation, the precision-recall across individual stages varied significantly, highlighting the uneven performance with the lowest precision at 0.54 in CS2 and recall in CS3 at the lowest at 0.56. In contrast, the QCVM classifications demonstrated more consistent performance, with the lowest precision of 0.66 in QCVM III and the lowest recall of 0.64. Therefore, the overall accuracy of the quantitative method was 0.78, indicating less variability and higher reliability than that of the qualitative method.

The quantitative method categorizes the CVM by ratios of body width to height, eschewing direct measurements. This approach stands in contrast to the qualitative method, which depends on the operator's visual inspection. The quantitative method utilizes multiple measurements, often with minimal differences between them, complicating the decision-making

process in QCVM staging. Consequently, deep learning models may facilitate more precise staging. Each method offers distinct advantages, and the choice between them should consider an additional skeletal index associated with growth maturity [25].

There is high agreement between CS1 and QCVM I, as well as between CS6 and QCVM VI, indicating that QCVM classifications align well with the Golden Standard, as described in Fig 4. However, the analysis also points out slight interactions where QCVM classifications do not align as well with the Golden Standard, particularly in CS3 with QCVM II and in CS4 with QCVM II and III. These discrepancies suggest that QCVM struggles with precision in intermediate stages, where the developmental characteristics may not be as distinctly defined as at the extremes. Consequently, the staging decision secondary indicator should be used alongside cervical vertebral maturation.

The AI model significantly outperforms human experts in terms of processing speed for cervical vertebral maturation assessments. The AI model processed and classified 360 images in just seven minutes, a fraction of the time required by human experts. This could lead to faster decision-making, improve patient management, and potentially reduce treatment times.

Finally, the study highlights the need for further research to enhance the clinical applicability of these methods. The absence of additional skeletal maturity indicators, such as wrist-hand radiographs or chronological age records, is a limitation that should be addressed in future studies. Incorporating these indicators could provide a more comprehensive assessment, further validating the reliability of the methods explored.

## 5. Conclusion

This study shows that incorporating AI into CVM assessments can greatly improve accuracy, efficiency, and reliability for orthodontists. In particular, compared to the AI-based qualitative CVM, the AI-based quantitative method stands out as a more precise tool for CVM evaluation. The fully automated QCVM model could give orthodontists a powerful tool to enhance cervical vertebral maturation staging.

## Supporting information

**S1 Table. Qualitative measurements of six maturation stages and orthodontic intervention timing.** (DOCX)

**S2 Table. Anatomical landmarks and their definitions used for QCVM analysis.** (DOCX)

**S3 Table. Quantitative measurement descriptions.** (DOCX)

## Acknowledgments

We sincerely appreciate Zhang Jia Yu and Fan Cheng Min from Chengdu Boltzmann Intelligence Technology Co., Ltd for their valuable assistance in the technical arrangement. We also extend our gratitude to our colleagues and research collaborators for their insightful discussions and continuous support.

## Author contributions

**Conceptualization:** Abbas Ahmed Abdulqader, Bushra Sufyan Almaqrami.

**Data curation:** Abbas Ahmed Abdulqader, Fulin Jiang, Yong Qiu.

**Formal analysis:** Bushra Sufyan Almaqrami.

**Investigation:** Abbas Ahmed Abdulqader.

**Methodology:** Abbas Ahmed Abdulqader, Fulin Jiang, Jinghong Yu, Yong Qiu, Juan Li.

**Project administration:** Juan Li.

**Software:** Fangyuan Cheng.

**Supervision:** Juan Li.

**Writing – original draft:** Abbas Ahmed Abdulqader.

**Writing – review & editing:** Jinghong Yu, Juan Li.

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
