## [Decision Letter · Decision Letter 0]

18 Mar 2025

PONE-D-25-06881Deep Learning Approaches for Quantitative and Qualitative Assessment of Cervical Vertebral Maturation Staging Systems.PLOS ONE

Dear Dr. Li,

Thank you for submitting your manuscript to PLOS ONE. After careful consideration, we feel that it has merit but does not fully meet PLOS ONE’s publication criteria as it currently stands. Therefore, we invite you to submit a revised version of the manuscript that addresses the points raised during the review process.

We look forward to receiving your revised manuscript.

Kind regards,

Dr. Sanjay Prasad Gupta, MDS- Orthodontics and Dentofacial Orthopedics

Academic Editor

PLOS ONE

Journal Requirements:

2**.** Please note that PLOS ONE has specific guidelines on code sharing for submissions in which author-generated code underpins the findings in the manuscript. In these cases, we expect all author-generated code to be made available without restrictions upon publication of the work. Please review our guidelines at https://journals.plos.org/plosone/s/materials-and-software-sharing#loc-sharing-code and ensure that your code is shared in a way that follows best practice and facilitates reproducibility and reuse.

3. In the online submission form, you indicated that “All data are available in the Orthodontics Department of West China Hospital of Stomatology at Sichuan University. The current study's datasets are available from the corresponding author upon request.”

6. Please ensure that you refer to Figure 5 in your text as, if accepted, production will need this reference to link the reader to the figure.

**Additional Editor Comments:**

The author’s work is appreciated but it would have been better if the authors incorporate the reviewer's suggestions to make this manuscript even better. Please follow the author guidelines in the revised manuscript.

Our decision for this manuscript is to revise (major revision).

Reviewers' comments:

Reviewer's Responses to Questions

**Comments to the Author**

1. Is the manuscript technically sound, and do the data support the conclusions?

Reviewer #1: Partly

Reviewer #2: Partly

2. Has the statistical analysis been performed appropriately and rigorously? 

Reviewer #1: No

Reviewer #2: I Don't Know

3. Have the authors made all data underlying the findings in their manuscript fully available?

Reviewer #1: Yes

Reviewer #2: Yes

4. Is the manuscript presented in an intelligible fashion and written in standard English?

Reviewer #1: Yes

Reviewer #2: No

5. Review Comments to the Author

Reviewer #1: Dear authors, I hope you will find these comments useful.

1. Authors should highlight the novelty of the study in the introduction.

2. It is unclear how many cephalograms were used for qualitative annotation, 6000 or 3600? “A qualitative visual classification of 6000 lateral cephalometric radiographs with unknown cervical vertebral maturation (CVM) stages was performed. The final data set consisted of 3600 radiographs and was even distributed at six classification stages.”

3. Did two orthodontists label all 3600 radiographs each in quantitative annotation? How was the ground truth established? The authors should make it explicit.

4. The conclusion should be backed by the findings, it is unclear whether the authors are referring to statistical significance. “This study demonstrates that the integration of AI in clinical orthodontic practice can significantly enhance the accuracy, efficiency, and reliability of cervical vertebral maturation assessments.”

Reviewer #2: Abstract:

• Aim: Keep the objective simple and avoid claims beyond the scope of the research. I would suggest avoiding the phrases such as “improving clinical workflows”, “patient outcomes”.

• Methodology: Remove the phrase “several statistical measures” and replace with the tools used.

• Result: mention the exact number (instead of less than 2mm)

• Conclusion: be specific to conclude findings from this study only. Only the second sentence seems to be suitable for conclusion. Others are just the opinion/recommendation from the authors.

Introduction:

• There was a recent systematic review on this topic published in BMC Oral health. It is worth citing this research in introduction section and highlighting the research gap that led to this study.

• https://bmcoralhealth.biomedcentral.com/articles/10.1186/s12903-025-05482-9

• The objectives of this study seem overpromising. Rather than “identifying the best approach for accurate and consistent CVM assessment”, this study is concentrated on finding whether AI generated staging correlates with human assessment or not.

Methods:

• Were any methods used to quantify inter-rater agreement?

Results:

• In tables 3 and 4, also present 95% CI along with mean values.

• The data from table 1 can be just described in one sentence and a table is not needed.

Discussion:

• The time to assess the staging directly appeared in this section without any information regarding it in the results section.

Conclusion:

• Conclude based on the results only. Do not express personal opinion here.

6. PLOS authors have the option to publish the peer review history of their article (what does this mean? ). If published, this will include your full peer review and any attached files.

**Do you want your identity to be public for this peer review?** For information about this choice, including consent withdrawal, please see our Privacy Policy .

Reviewer #1: No

Reviewer #2: **Yes**

---

## [Author Response · Author response to Decision Letter 1]

8 Apr 2025

Dear Dr. Sanjay Prasad Gupta,

Thank you for your detailed feedback on our submission. We have carefully reviewed and addressed each of the points raised in accordance with PLOS ONE’s guidelines:

1. Manuscript Style and File Naming

We have revised our manuscript to align with PLOS ONE’s formatting and style requirements, including appropriate file naming conventions for all uploaded files.

2. Code Sharing

We confirm that the QCVM and CVM-generated code used in our study has been made publicly available without restriction. The code is referenced in the Data Availability Statement and is hosted at https://github.com/ortho2024/QCVM-CVM. We have ensured the repository includes documentation and versioning to support reproducibility and reuse.

3. Data Availability Statement

We have revised the data availability statement for clarity and compliance. The updated statement in the manuscript reads:

“We have made all the code available at “https://github.com/ortho2024/QCVM-CVM” that can be used to replicate this study. The data underlying the findings of this study cannot be made publicly available due to ethical and legal restrictions. The datasets can't be public deposition because it would compromise patient privacy and violate the protocol approved by the research ethics board of West China Hospital of Stomatology, Sichuan University. However, the data are available from the corresponding author upon reasonable request, subject to approval by the ethics committee and compliance with data protection regulations. Researchers interested in accessing the data may contact [Juan Li] at [E-mail: lijuan@scu.edu.cn] to submit their request”

4. Ethics Statement Placement

The ethics statement has been relocated to the Methods section of the manuscript and removed from any other section, as per your instructions.

5. Figure Captions

We have ensured that each figure is cited in the text at its first mention, includes a complete caption within the manuscript, and has been separately uploaded as a high-quality TIFF file.

6. Reference to Figure 5

We have added an explicit reference to Figure 5 within the main text.

7. Supporting Information Captions and Citations

Captions for all Supporting Information files have been included at the end of the manuscript, and all corresponding in-text citations have been updated accordingly.

Response to Reviewer #1

We sincerely thank the reviewer for the thoughtful and constructive comments. We have addressed each point in detail below and revised the manuscript accordingly. And the changed sentences are marked in “Red”.

1. Authors should highlight the novelty of the study in the introduction.

Response: We appreciate the reviewer’s suggestion. We have revised the Introduction section to show our study contribution explicitly. “In this study, we investigate the potential of deep learning to enhance the accuracy and efficiency of CVM assessment. We conduct a comparative analysis of qualitative CVM and quantitative QCVM methods when augmented with AI. Thereby supporting better-informed decisions on the timing of orthopedic treatment.

Our main contributions are as follows:

• We present a large, evenly distributed dataset encompassing all stages of cervical vertebral maturation, collected from multiple centers to ensure diversity and generalizability.

• As the first comparative study of AI based on QCVM and CVM, this work provides critical insights into the effectiveness and clinical practicality of various cervical maturation staging methods.

• We developed deep learning models by employing six maturation stages assessment methods for both quantitative and qualitative analysis (as depicted in Figure 1). The CVM staging adheres to the original framework proposed by Baccetti et al.[8] This integration allows to bridge classical methodologies with modern AI-driven approaches effectively.”

2. It is unclear how many cephalograms were used for qualitative annotation, 6000 or 3600?

Response: Thank you for pointing this out. We have revised the Methods section for clarity.

“An initial dataset of 6000 lateral cephalometric radiographs was randomly collected from six orthodontic clinics. After applying exclusion criteria, 2400 cephalograms were excluded, resulting in a final dataset of 3600 images. Exclusions were based on the following criteria: (1) incomplete visualization of all three cervical vertebrae, (2) balancing the sample distribution across Cervical Stage (CS) classifications (CS1 to CS6) to ensure equal representation, and (3) standardizing the dataset to maintain uniform sample sizes across all stages for consistent analysis”

3. Did two orthodontists label all 3600 radiographs each in quantitative annotation? How was the ground truth established? The authors should make it explicit.

Response: Thank you for highlighting the need for clarification. We have updated the Methods section to specify that two experienced orthodontists annotated all 3600 cephalograms by identifying cervical vertebral landmarks. The consensus-based annotations served as the ground truth for training and evaluating the QCVM landmark detection system. These landmarks were then used to compute QCVM measurements, which were subsequently mapped to CVM stages. To automate this stage classification, we employed a CatBoost model that extracted relevant features from the corresponding cephalograms, which had been qualitatively evaluated based on the McNamara user guide. These clarifications have been incorporated into the manuscript in Section 2.2(b) Quantitative Method (Landmark Annotation) and Section 2.3(c) Quantitative cervical vertebral maturation staging. “paste”

4. The conclusion should be backed by the findings; it is unclear whether the authors are referring to statistical significance. “This study demonstrates that the integration of AI in clinical orthodontic practice can significantly enhance the accuracy, efficiency, and reliability of cervical vertebral maturation assessments.”

Response: Thank you for this valuable comment. We have revised the Conclusion to reflect and align more accurately with the findings presented in the manuscript. Specifically, we have clarified the basis of our claims by emphasizing the demonstrated improvements in accuracy, efficiency, and reliability, supported by the performance metrics of the proposed model. The updated conclusion now reads: “This study shows that incorporating AI into CVM assessments can greatly improve accuracy, efficiency, and reliability for orthodontists. In particular, compared to the AI-based qualitative CVM, the AI-based quantitative method stands out as a more precise tool for CVM evaluation. The fully automated QCVM model could give orthodontists a powerful tool to enhance cervical vertebral maturation staging.”.

Response to Reviewer #2

We would like to thank Reviewer #2 for the thoughtful and constructive feedback, which has helped us improve the clarity and quality of the manuscript. Below, we provide our detailed responses to each comment and describe the corresponding revisions made. And the changed sentences are marked in “Red”.

1. Abstract:

• Aim: Keep the objective simple and avoid claims beyond the scope of the research. I would suggest avoiding the phrases such as “improving clinical workflows”, “patient outcomes”.

• Methodology: Remove the phrase “several statistical measures” and replace with the tools used.

• Result: mention the exact number (instead of less than 2mm)

• Conclusion: be specific to conclude findings from this study only. Only the second sentence seems to be suitable for conclusion. Others are just the opinion/recommendation from the authors.

Response:

Thank you for these helpful suggestions. We have revised the Abstract to adhere to the PLOS ONE format and have carefully addressed each of your points:

The objective has been simplified and now focuses only on the purpose of comparing AI-based qualitative and quantitative methods for CVM staging. “To investigate the potential of artificial intelligence (AI) in Cervical Vertebral Maturation (CVM) staging, we developed and compared AI-based qualitative CVM and AI-based quantitative QCVM methods.”

The phrase “several statistical measures” has been replaced with the exact tools used: Pearson correlation coefficient, mean square error (MSE), success detection rate (SDR), precision-recall metrics, and F1 score.

The results now specify the mean annotation difference as 0.17–0.55 mm rather than saying “less than 2 mm” and changed as follows: “Statistical analyses evaluated the methods' performance, including the Pearson correlation coefficient, mean square error (MSE), success detection rate (SDR), precision-recall metrics, and the F1 score.”

The conclusion has been revised to reflect findings from this study only and avoids general recommendations. “These findings suggest that the AI-based quantitative QCVM method offers superior performance, with higher agreement rates and classification accuracy compared to the AI-based qualitative CVM approach, indicating the fully automated QCVM model could give orthodontists a powerful tool to enhance cervical vertebral maturation staging.”

2. Introduction

• There was a recent systematic review on this topic published in BMC Oral Health. It is worth citing this research in the introduction section and highlighting the research gap that led to this study. https://bmcoralhealth.biomedcentral.com/articles/10.1186/s12903-025-05482-9

• The objectives of this study seem overpromising. Rather than “identifying the best approach for accurate and consistent CVM assessment”, this study is concentrated on finding whether AI-generated staging correlates with human assessment or not.

Response:

Thank you for highlighting this. We have cited the recent systematic review published in BMC Oral Health and used it to support the rationale for our study: “Additionally, a meta-analysis systematically evaluates the performance of artificial intelligence (AI) models in assessing cervical vertebral maturation (CVM) from radiographs, highlighting their potential to enhance accuracy and reliability compared to traditional clinical methods[17].”

Additionally, we have revised the study objectives in the Introduction to focus on evaluating the correlation between AI-generated staging and expert assessment, rather than making definitive claims about the best method: “In this study, we investigate the potential of deep learning to enhance the accuracy and efficiency of CVM assessment. We conduct a comparative analysis of qualitative CVM and quantitative QCVM methods when augmented with AI. Thereby supporting better-informed decisions on the timing of orthopedic treatment.

Our main contributions are as follows:

• We present a large, evenly distributed dataset encompassing all stages of cervical vertebral maturation, collected from multiple centers to ensure diversity and generalizability.

• As the first comparative study of AI based on QCVM and CVM, this work provides critical insights into the effectiveness and clinical practicality of various cervical maturation staging methods.

• We developed deep learning models by employing six maturation stages assessment methods for both quantitative and qualitative analysis (as depicted in Figure 1). The CVM staging adheres to the original framework proposed by Baccetti et al.[8] This integration allows to bridge classical methodologies with modern AI-driven approaches effectively.”

3. Methods:

• Were any methods used to quantify inter-rater agreement?

Response: Thank you for your reminder. We have expanded the Methods section to include the inter-rater agreement assessment. Specifically:

“To evaluate agreement among the three annotators in qualitative CVM assessment, we calculated Cohen’s Kappa and Fleiss’ Kappa. A 10% random sample from each CVM stage was used. The results showed that Cohen’s Kappa between the senior orthodontist and Junior Orthodontist 1 was 0.78, with Junior Orthodontist 2 was 0.84, and between the two juniors was 0.85. The overall Fleiss’ Kappa was 0.82, indicating high inter-rater reliability.

For landmark annotation reliability, we evaluated 100 radiographs annotated independently by two junior orthodontists. The weighted SDR across 13 landmarks was 97% at a 1 mm threshold, confirming consistent landmark placement.”

4. Results:

• In tables 3 and 4, also present 95% CI along with mean values.

• The data from table 1 can be just described in one sentence and a table is not needed.

Response:

We appreciate these suggestions. As recommended, we changed the tables and refreshed the table citation numbers.

Table 1 has been removed, and the content is now described briefly in the Methods section.

Tables 2 and 4 (original Table 3 and 4) have been revised to include 95% confidence intervals alongside mean values.

Additionally, we included a new table (Tabel 3) presenting SDR values at 0.5 mm, 1.0 mm, and 1.5 mm thresholds for greater clarity.

Landmarks Mean SD Median Min Max 95%CI

C2p (mm) 0.55 0.37 0.50 0 1.91 0.51 ~ 0.59

C2m (mm) 0.41 0.33 0.35 0 1.42 0.38 ~ 0.45

C2a (mm) 0.51 0.42 0.40 0 2.43 0.46 ~ 0.55

C3up (mm) 0.29 0.29 0.20 0 1.89 0.26 ~ 0.32

C3ua (mm) 0.20 0.19 0.19 0 1.55 0.18 ~ 0.22

C3lp (mm) 0.32 0.29 0.26 0 1.59 0.29 ~ 0.35

C3la (mm) 0.29 0.29 0.21 0 2.06 0.26 ~ 0.32

C3m (mm) 0.17 0.18 0.13 0 1.25 0.32 ~ 0.39

C4up (mm) 0.35 0.32 0.26 0 1.47 0.13 ~ 0.17

C4ua (mm) 0.21 0.21 0.192 0 1.07 0.21 ~ 0.26

C4um (mm) 0.15 0.17 0.12 0 1.57 0.30 ~ 0.36

C4lp (mm) 0.33 0.28 0.27 0 2.09 0.26 ~ 0.32

C4la (mm) 0.29 0.281 0.21 0 2.37 0.15 ~ 0.19

C4m (mm) 0.21 0.19 0.18 0 1.06 0.18 ~ 0.23

Table 2 Positional difference of landmarks between human and AI. A value of 0 indicates perfect agreement, while values less than 2 mm are considered clinically acceptable. CL (Confidence Interval) represents the 95% confidence range around the mean value, indicating the reliability of the estimate.

Landmarks SDR

≤ 0.5 mm (%) ≤ 1.0 mm (%) ≤ 1.5 mm (%)

C2p 49.86 87.19 98.33

C2d 64.35 93.87 100.00

C2a 57.94 85.52 97.21

C3up 79.39 97.49 99.72

C3ua 91.64 99.72 99.72

C3lp 75.49 97.77 99.44

C3la 79.94 96.94 99.44

C4up 70.75 93.87 100.00

C4um 95.54 99.72 99.72

C4am 87.19 99.44 100.00

C4lp 73.82 98.89 99.72

C4la 80.50 98.33 99.44

C3ld 93.59 99.72 100.00

C4ld 92.20 99.72 100.00

C3pm 86.91 98.89 100.00

C3am 90.81 99.72 100.00

C3um 97.49 100.00 100.00

C4pm 87.47 99.16 100.00

C4ua 91.36 99.72 100.00

Table 3: Success Detection Rate (SDR) – the percentage of AI predictions within the given distance from human landmarks.

Measurement AI 　 Orthodontists Pearson Correlation MSE

Mean Sd 95%CI Mean Sd 95%CI

C2Conc mm 0.64 0.59 0.58 ~ 0.70 　 0.74 0.58 0.80 ~ 0.73 0.73 0.19

C3Conc mm 0.7 0.64 0.63 ~ 0.76 0.64 0.6 0.71 ~ 0.93 0.93 0.058

C4Conc mm 0.58 0.54 0.52 ~ 0.63 0.58 0.53 0.63 ~ 0.9 0.9 0.054

C3PAR (ratio) 1.34 0.18 1.32 ~ 1.36 1.33 0.18 1.35 ~ 0.93 0.93 0.004

C3BAR (ratio) 1.68 0.48 1.63 ~ 1.73 1.68 0.47 1.73 ~ 0.98 0.98 0.006

C4PAR (ratio) 1.4 0.16 1.38 ~ 1.41 1.4 0.19 1.42 ~ 0.88 0.88 0.007

C4BAR (ratio) 1.71 0.44 1.66 ~ 1.76 　 1.73 0.45 1.78 ~ 0.97 0.97 0.01

Table 4 Comparison of quantitative measurements of orthodontists and deep learning

5. Discussion:

• The time to assess the staging directly appeared in this section without any information regarding it in the results section.

Response:

Thank you for pointing this out. We have revised the Results section to include a subsection titled “Efficiency Comparison”, which presents the average time taken by the AI system versus human raters for CVM staging. This change has been reflected in the Discussion section accordingly. The changed paragraph showed as follows:

“Efficiency Comparison: The timing records of 360 test data between expert classification and the trained model revealed that the AI model processed and classified the images in just 7 minutes for both CVMS and QCVM, whereas junior orthodontists took an average of 53 minutes in CVM assessment and 347.64 minutes in landmark localization of QCVM. This significant difference in processing time suggests that the AI model can offer much faster diagnostic tim

---

## [Editor Report · Decision Letter 1]

15 Apr 2025

Deep Learning Approaches for Quantitative and Qualitative Assessment of Cervical Vertebral Maturation Staging Systems.

PONE-D-25-06881R1

Dear Dr. Li,

We’re pleased to inform you that your manuscript has been judged scientifically suitable for publication and will be formally accepted for publication once it meets all outstanding technical requirements.

Kind regards,

Dr. Sanjay Prasad Gupta, MDS- Orthodontics and Dentofacial Orthopedics

Academic Editor

PLOS ONE

---

## [Editor Report · Acceptance letter]

PONE-D-25-06881R1

PLOS ONE

Dear Dr. Li,

I'm pleased to inform you that your manuscript has been deemed suitable for publication in PLOS ONE. Congratulations! Your manuscript is now being handed over to our production team.

Kind regards,

on behalf of

Dr. Sanjay Prasad Gupta

Academic Editor

PLOS ONE